# PGC-1α and MEF2 Regulate the Transcription of the Carnitine Transporter OCTN2 Gene in C2C12 Cells and in Mouse Skeletal Muscle

**DOI:** 10.3390/ijms232012304

**Published:** 2022-10-14

**Authors:** Katerina Novakova, Michael Török, Miljenko Panajatovic, Jamal Bouitbir, François H. T. Duong, Christoph Handschin, Stephan Krähenbühl

**Affiliations:** 1Division of Clinical Pharmacology & Toxicology, University Hospital, 4031 Basel, Switzerland; 2Molecular & Systems Toxicology, Department of Pharmaceutical Sciences, University of Basel, 4001 Basel, Switzerland; 3Department of Biomedicine, University of Basel, 4001 Basel, Switzerland; 4Biocenter, University of Basel, 4001 Basel, Switzerland

**Keywords:** carnitine, OCTN2, SLC22A5 gene transcription, MEF2, PGC-1α, p38 MAPK

## Abstract

OCTN2 (SLC22A5) is a carnitine transporter whose main function is the active transport of carnitine into cells. In skeletal muscle and other organs, the regulation of the SLC22A5 gene transcription has been shown to depend on the nuclear transcription factor PPAR-α. Due to the observation that the muscle OCTN2 mRNA level is maintained in PPAR-α knock-out mice and that PGC-1α overexpression in C2C12 myoblasts increases OCTN2 mRNA expression, we suspected additional regulatory pathways for SLC22A5 gene transcription. Indeed, we detected several binding sites of the myocyte-enhancing factor MEF2 in the upstream region of the SLC22A5 gene, and MEF2C/MEF2D stimulated the activity of the OCTN2 promoter in gene reporter assays. This stimulation was increased by PGC-1α and was blunted for a SLC22A5 promoter fragment with a mutated MEF2 binding site. Further, we demonstrated the specific binding of MEF2 to the SLC22A5 gene promoter, and a supershift of the MEF2/DNA complex in electrophoretic mobility shift assays. In immunoprecipitation experiments, we could demonstrate the interaction between PGC-1α and MEF2. In addition, SB203580, a specific inhibitor of p38 MAPK, blocked and interferon-γ stimulated the transcriptional activity of the SLC22A5 gene promoter. Finally, mice with muscle-specific overexpression of OCTN2 showed an increase in OCTN2 mRNA and protein expression in skeletal muscle. In conclusion, we detected and characterized a second stimulatory pathway of SLC22A5 gene transcription in skeletal muscle, which involves the nuclear transcription factor MEF2 and co-stimulation by PGC-1α and which is controlled by the p38 MAPK signaling cascade.

## 1. Introduction

Carnitine (3-hydroxy-4-(trimethylammonio)butyrate, in the entire manuscript standing for L-carnitine) is a polar molecule that is essential for long-chain fatty acid transport into the mitochondrial matrix, where fatty acids can undergo β-oxidation [1,2,3]. The carnitine body stores are maintained mainly by the oral ingestion of nutrients containing carnitine (such as meat). In addition, in mammals, carnitine can also be biosynthesized from lysine [1,4]. Most carnitine is contained in skeletal muscle, where the concentration is in the millimolar range [5,6]. In comparison, the plasma carnitine plasma concentration is in the range of 30 µM [5], indicating that the transport of carnitine from the plasma into tissues such as skeletal muscle must be active. OCTN2 (SLC22A5) is the most important carnitine transporter with a high expression mainly in the kidney, but also in other organs [7,8]. OCTN2 is located in the plasma membrane of most cells and transports carnitine into cells in a sodium-dependent fashion [9,10,11]. The physiological importance of OCTN2 and of carnitine is demonstrated in mice and humans with loss of function mutations of the SLC22A5 gene, which leads to systemic carnitine deficiency due to renal losses of carnitine [12,13]. Systemic carnitine deficiency is associated with an impaired fatty acid metabolism, causing hypoketotic hypoglycemia in affected patients during starvation, which may be lethal [14].

Considering the essential role of OCTN2 in carnitine tissue distribution and renal reabsorption, understanding the regulation of OCTN2 expression is important. Van Vlies et al. have shown that OCTN2 mRNA expression is decreased in the livers and hearts of PPAR-α knock-out compared to wild-type mice, but not in the kidney or skeletal muscle [15]. The role of PPAR-α in stimulating the expression of OCTN2 was subsequently confirmed in rat liver [16], pig liver, skeletal muscle, and the small intestine [17], as well as in mouse liver, the small intestine, and kidney and skeletal muscle [18]. Interestingly, the strong PPAR-α response element is not located in the precoding region of the SLC22A5 gene, but in its first intron [19,20,21]. Depending on the tissue or cell type, additional regulation mechanisms for OCTN2 expression have been described, e.g., osmotic stress for epididymal cells [22], decrease in the plasma (and proximal tubular fluid) carnitine concentration for the renal expression of OCTN2 [23,24], the stimulation of PPAR-γ for colonic OCTN2 expression [19], and PPAR-β/δ stimulation in MDBK cells (a bovine kidney cell line) [25].

While the role of PPAR-α for the stimulation of OCTN2 expression is clearly established for the liver, it is less clear for skeletal muscle. In PPAR-α knock-out mice, skeletal muscle OCTN2 expression was not lower than in wild-type mice [15,18], and the increase in OCTN2 mRNA expression by PPAR-α activators was lower in skeletal muscle than in the liver [17,18]. These findings suggest that additional factors may be involved in the regulation of OCTN2 expression in skeletal muscle. One of these factors may be the peroxisome proliferator-activated receptor-gamma coactivator (PGC-1α). PGC-1α is a protein with a strong expression in tissues with a high energy demand such as skeletal muscle where it acts as a transcriptional regulator of a large number of proteins involved in many metabolic pathways. In skeletal muscle, PGC-1α stimulates mitochondrial biogenesis [26], the switch from fast-twitch (glycolytic) into slow-twitch, mitochondria-rich, oxidative muscle fibers [27], as well as the expression of enzymes involved in fatty acid transport (fatty acid translocase/CD36), carnitine palmitoyltransferase (CPT) I, and medium-chain acyl-CoA dehydrogenase [28]. Since PGC-1α is a co-activator of PPAR-α [29] and PPAR-α stimulates OCTN2 expression in skeletal muscle [17,18], overexpression of PGC-1α can be expected to stimulate OCTN2 expression. However, PGC-1α may stimulate OCTN2 expression in skeletal muscle by interaction with additional transcription factors associated with muscle growth. One of these candidate proteins is myocyte enhancer factor 2 (MEF2), which is known to interact with PGC-1α [30,31] and which has an important role in skeletal muscle differentiation and growth [32]. Based on these considerations, we decided to study the effect of PGC-1α and MEF2 on the transcriptional activity of the human SLC22A5 gene precoding region cloned into C2C12 myoblasts.

## 2. Results

The aim of this project was to investigate the potential role of PGC-1α and associated proteins in the regulation of the transcription of the SLC22A5 gene. Previous studies have shown that PPAR-α stimulation increases OCTN2 mRNA expression in the skeletal muscle of mice and pigs [17,18]. However, since PPAR-α knock-out mice showed no significant decrease in skeletal muscle OCTN2 mRNA expression [15,18], we investigated the possibility of additional regulators of skeletal muscle SLC22A5 gene transcription.

To investigate a possible role of PGC-1α in SLC22A5 gene transcription, we first determined the expression of OCTN2 mRNA and protein in C2C12 myoblasts overexpressing PGC-1α. As shown in Figure 1, overexpression of PGC-1α was associated with an approximately four-fold increase in OCTN2 mRNA expression (Figure 1A) and a doubling in OCTN2 protein expression (Figure 1B). In agreement with mRNA and protein expression, the transport of ^3^H-carnitine into C2C12 cells overexpressing PGC-1α was approximately three-fold higher compared to C2C12 control cells transfected with the empty vector (Figure 1C).

Although PGC-1α could act via stimulation of PPAR-α, we were searching for additional transcription factors whose activities could be stimulated by PGC-1α. A possible candidate was MEF2, whose activity has been described to be dependent, at least partially, on PGC-1α [30,31]. Indeed, as shown in Figure 2, we found several MEF2 binding sites in the SLC22A5 gene 5′ upstream region, suggesting that MEF2 could be involved in the regulation of SLC22A5 gene transcription.

To assess the function of the MEF2 binding sites, we produced different 5′-truncated fragments of the SLC22A5 gene 5′ upstream region and inserted them into the pGL3 vector. After the transfection of C2C12 myoblasts with these plasmids, we assessed the functionality of the transfected fragments using a dual luciferase assay. As shown in Figure 3A, the −1962/+244 (2206 bp) and the −3256/+244 (3500 bp) promoter segments were associated with the highest reporter gene activity. These segments contained several MEF2 binding sites (e.g., −820/−803, −902/−884, and −1614/−1597, as shown in Figure 2), supporting the notion that MEF2 increases the transcription of the SLC22A5 gene. Furthermore, for the 2206 bp and the 3500 bp fragments, the reporter gene activity was higher in C2C12 myoblasts overexpressing PGC-1α, suggesting an interaction between PGC-1α and MEF2.

To prove further the role of MEF2 and PGC-1α in SLC22A5 gene transcription, we transfected C2C12 myoblasts with the −1962/+244 promoter segment (Figure 3B) and co-transfected the myoblasts with MEF2C/D and/or PGC-1α. Co-transfection with MEF2D, MEF2D, or PGC-1α numerically increased the activity of the promoter, and the combination of MEF2C/D and PGC-1α overexpression was associated with a statistically significant increase in the promoter activity.

As a further proof of the role of MEF2 in the activation of SLC22A5 gene transcription, we prepared the −1624/−1595 fragment of the precoding region of the SLC22A5 gene, which carries one MEF2 binding site (Figure 2). We introduced a T to G mutation in position −1607 and tested the activity of the non-mutated and the mutated fragments using the dual luciferase reporter gene assay in C2C12 cells overexpressing MEF2D. As shown in Figure 4A, MEF2D caused an activation of the reporter gene only in the presence of the non-mutated fragment, whereas the T to G mutation eliminated the stimulatory effect of MEF2D.

To show the physical interaction of MEF2 with the regulatory sequence −1624/−1595, we performed gel-shift competition and electrophoretic mobility shift (EMSA) assays. As shown in Figure 4B, we could prevent the binding of MEF2 to the labeled DNA probe by adding the non-labeled probe, whereas the mutated DNA probe did not show this interaction. Furthermore, we observed a supershift of the MEF2-DNA complex when we added an anti-MEF2 antibody. In contrast, a supershift was not observed when an anti-PGC-1α antibody or control IgG was added to the MEF2-DNA complex.

Since we had used nuclear extracts not containing PGC-1α in the EMSA assay, we could have missed an interaction between PGC-1α and MEF2. Therefore, we investigated a possible interaction between PGC-1α and MEF2 using immunoprecipitation experiments. We incubated C2C12 myoblast lysates with anti-MEF2 antibodies, separated the precipitate with gel electrophoresis, and assayed the blot using anti-MEF2 and anti-PGC-1α antibodies. Since interferon-γ can activate p38 MAP kinase (p38 MAPK) [33,34], which activates MEF2 [30,32], some incubations were previously treated with interferon-γ for 12 h. As shown in Figure 5A, immunoprecipitation proved the interaction between MEF2 and PGC-1α. In addition, treatment with interferon-γ was associated with an increase in PGC-1 in the anti-MEF2 precipitate (Figure 5B), suggesting that stimulation of p38 MAPK signaling increases the interaction between MEF2 and PGC-1α.

To further investigate the stimulation of the SLC22A5 gene transcription via the interferon-γ/P38 MAPK/MEF2D axis, we inserted the SLC22A5 gene promoter into C2C12 myoblasts and analyzed the reporter gene activity using the dual luciferase reporter assay in the absence and presence of interferon-γ. As shown in Figure 6A, in C2C12 cells overexpressing PGC-1α and MEF2D, interferon-γ increased the activity of the SLC22A5 gene promoter by a factor of approximately 2.3. Using the same cell system, we could block the reporter gene activity by adding the P38 MAP kinase blocker SB203580 (Figure 6B). These experiments suggest that the p38 MAPK pathway, which can be activated by interferon-γ, is involved in the regulation of SLC22A5 gene transcription via stimulation of MEF2.

To investigate the role of PGC-1α in the SLC22A5 gene transcription in vivo, we quantified OCTN2 mRNA and OCTN2 protein expression in different organs of wild-type mice (WT mice), mice with muscle-specific PGC-1α overexpression (OE mice), and PGC-1α knock-out (KO mice). As shown in Figure 7, compared to WT mice, the mRNA expression of OCTN2 was higher in the red and white gastrocnemius muscle of PGC-1α OE mice compared to WT mice, confirming that PGC-1α stimulates OCTN2 expression in mouse skeletal muscle. In addition, in support of the role of PGC-1α in OCTN2 expression, OE mice showed a significant increase in gastrocnemius OCTN2 protein expression compared to WT and KO mice.

## 3. Discussion

The observation that PPAR-α knock-out mice do not show a significant decrease in skeletal muscle OCTN2 expression [15,18] prompted us to investigate the existence of additional regulators of the SLC22A5 gene transcription in skeletal muscle. PPAR-α has been identified as a nuclear transcription factor for the SLC22A5 gene in different animals and tissues [16,17,18,35].

Regarding the important role of PGC-1α in the transcription of genes related to energy metabolism [26,28], we first investigated the effect of PGC-1α overexpression on OCTN2 mRNA and protein abundance, as well as OCTN2 function in C2C12 myoblasts. The observed increase in OCTN2 mRNA and expression and OCTN2 function could be expected since PGC-1α is a co-regulator and activator of PPAR-α [29]. However, PGC-1α interacts with many other transcription factors [28,29], among them the proteins of the MEF2 family [30,31]. MEF2 proteins are well-characterized nuclear transcription factors with a high expression in skeletal muscle and the central nervous system [32]. Since MEF2 proteins have been shown to stimulate the differentiation of myocytes in skeletal muscle and the heart [30,32,36] and to interact with PGC-1α [30,31], they were good candidates to mediate the effect of PGC-1α in C2C12 cells. Indeed, the investigation of the precoding region of the SLC22A5 gene revealed several binding motifs for MEF2 proteins. We confirmed the interaction of MEF2 with these binding sites using gel-shift and electrophoretic mobility shift assays and the activation of the SLC22A5 gene promoter by MEF2C and MEF2D using reporter gene assays. These assays also revealed the interaction of MEF2 with PGC-1α, which was corroborated by immunoprecipitation experiments.

The activation and expression of MEF2 proteins are regulated by different signaling pathways, among them the MAP kinase pathway [32]. We could show that the MAPK inhibitor SB203580 blocks the PGC-1α/MEF2-induced activity of the SLC22A5 gene promoter, suggesting that the MAPK pathway is involved in the PGC-1α/MEF2 stimulation of the SLC22A5 gene promoter activity. This result was confirmed by the finding that interferon-γ, which is a stimulator of p38 MAPK [33,34], increased the transcriptional activity of the SLC22A5 gene promoter. Importantly, treatment of HEK293 or Caco2BBE cells with interferon-γ has been shown to be associated with an increase in OCTN2 expression and function, supporting our findings regarding the effect of that interferon-γ [37,38]. Furthermore, overexpression of PGC-1α in the skeletal muscles of mice caused an increase in skeletal muscle OCTN2 mRNA and protein levels, confirming the stimulation of the SLC22A5 gene transcription by PGC-1α. Interestingly, PGC-1α knock-out was associated with only a numerical decrease in skeletal muscle OCTN2 mRNA and protein expression (Figure 7). This finding suggests the existence of additional regulators of SLC22A5 gene transcription in skeletal muscle and/or that the presence of PPAR-α and MEF2 in the absence of PGC-1α is sufficient to maintain SLC22A5 gene transcription.

Carnitine is essential for the transport of long-chain fatty acids into the mitochondrial matrix [1,2,3], where fatty acids are metabolized for energy production. The essential role of carnitine is demonstrated in animals and humans with a reduced function of OCTN2, which leads to systemic carnitine deficiency [12,13,14]. Skeletal muscle and cardiac dysfunction are important symptoms of systemic carnitine deficiency. Regarding these important functions of carnitine and considering that carnitine must be transported actively into skeletal muscle and not by diffusion, it is not astonishing that the transcription of the SLC22A5 gene is regulated by different pathways. The finding that PGC-1α interacts with both MEF2 and PPAR-α underlines the important role of PGC-1α in the regulation of the expression of proteins involved in energy metabolism.

In conclusion, we could provide evidence for a second pathway of the stimulation of the SLC22A5 gene transcription in addition to PPAR-α in skeletal muscle, which involves MEF2 as a nuclear transcription factor and co-stimulation by PGC-1α. The finding that the skeletal muscle SLC22A5 gene transcription is stimulated by different pathways underscores the essential role of carnitine in skeletal muscle energy metabolism.

## 4. Materials and Methods

### 4.1. Cells and Plasmids

Human liver carcinoma cell line HepG2, the murine skeletal muscle cell line C2C12, and human embryonic kidney (HEK293T) cells were obtained from the American Type Culture Collection (ATCC, LGC Standards, Molsheim, France). The luciferase reporter gene plasmid pGL3-enhancer and pGL3-control, and Renilla luciferase internal control plasmid pRL-TK, were purchased from the Promega Corporation (Promega, Madison, WI, USA). Pc DNA-PGC1α and pc DNA-MEF2D were purchased from Addgene (Watertown, MA, USA). The pCR2.1-TOPO plasmid was from Invitrogen (Basel, Switzerland). The plasmids pMD2.G (Addgene #12259), pCMVR8.74 (Addgene #22036), and pCW57-MCS1-P2A-MCS2 (Addgene #80921) were from Addgene (Watertown, MA, USA).

### 4.2. Enzymes and Antibodies

AccuTaq LA DNA Polymerase was from Sigma (Sigma-Aldrich, St. Louis, MO, USA), the T4 DNA Ligase from Promega (Promega, Switzerland), T4 Polynucleotide Kinase, Shrimp Alkaline Phosphatese, restriction enzymes Acc65I, AccI, SmaI, Hind II, XhoI, NheI, Cfr9I, XbaI, XmaJi, Cfr10I, Eco32I from Fermentas (Fermentas, Payerne, Switzerland), Taq-Polymerase and LipofectaminTM 2000 from Invitrogen (Invitrogen, Switzerland), PfuUltra Hotstar DNA Polymerase from Agilent (Agilent, Basel, Switzerland), and the Big dye Terminator from Applied Biosystems (Applied Biosystems, Muttenz, Switzerland). The p-38 MAPK inhibitor (SB203580) was obtained from Jena Bioscience (Jena Bioscience GmbH, Jena, Germany). Antibodies against β-actin (sc-47778, 1:1000), PGC-1α (sc-518025, 1:1000), and MEF2 (1:1000) were from Santa Cruz Biotechnology (Dallas, TX, USA), against OCTN2 (ab180757, 1:1000) from Abcam (Cambridge, UK) and the control IgG from Santa Cruz Biotechnology (Dallas, TX, USA).

The DNA purification kit NucleoSpin^®^Tissue and the NucleoBond^®^AX Plasmid Purification kit were from Macherey-Nagel (Düren, Germany), the QIAquick Gel Extraction Kit from Qiagen (Hilden, Germany), the Dual-Luciferase^®^ Reporter 1000 Assay System from Promega (Promega, Switzerland), and Micro Bio-spin chromatography columns and the Bio-Rad protein assay from Bio-Rad (Bio-Rad Laboratories AG, Cressier, Switzerland).

### 4.3. Lentivirus Production and C2C12 Cell Transduction

Production of replicant deficient lentivirus was done by transient transfection of HEK293T cells with plasmids pMD2.G, pCMVR8.74, and the pCW57-MCS1-P2A-MCS2 transfer plasmid with the inserted Ppargc1a mouse gene, as previously described [39]. Plasmid pCW57-MCS1-P2A-MCS2 was a gift from Adam Karpf (University of Nebraska, Lincoln, NE, USA) and plasmids pMD2.G and pCMVR8.74 were a gift from Didier Trono (École Polytechnique Fédérale de Lausanne, Lausanne, Switzerland). The HEK293T cells were transfected with plasmids for lentiviral production using the Metafectene^®^ Pro (Biontex Laboratories, Munich, Germany), according to the manufacturer’s instructions. Viral products were collected in the cell supernatant from the growth medium of transfected HEK293T cells after 48 h and passed through a 0.2 μm filter. Functional titration was performed by transduction of C2C12 myoblasts with a serially diluted virus in the presence of polybrene (5 μg/mL, Merck KGaA, Darmstadt, Germany) for 6 h followed by blasticidin (Merck KGaA, Darmstadt, Germany) selection 48 h post-infection. PGC-1α overexpression in transduced C2C12 cells was obtained by supplementation of the cell culture medium with 20 ng/mL doxycycline (D3072-1ML, Merck KGaA, Darmstadt, Germany) for 2 weeks.

### 4.4. Bioinformatic Analysis of the Upstream Sequence of the SLC22A5 Gene Promoter

The upstream region of the SLC22A5 gene was searched using the NCBI genome database. The transcription factor binding sites were analyzed using MatInspector (Genomatix, Munich, Germany).

### 4.5. PCR Primer Design and Isolation of the SLC22A5 Gene Promoter

Human genomic DNA was extracted from HepG2 cells using a genomic DNA extraction kit (Qiagen, Hilden, Germany) and was used as a template. In accordance with the DNA sequence of the SLC22A5 gene regulatory region and its restriction enzyme map, as well as the structural features of the target DNA fragments and the expression vector pGL3 plasmid, PCR amplification primers were designed (forward primer 5′-GCTGCTTATAACATGACAGCTTGCTTGTCT-3′ and reverse primer 5′-GTCACCTCGTCGTAGTCCCGCAT-3′). Using these primers, we could amplify the human SLC22A5 gene promoter fragment −3256/+244. The reaction settings were pre-denaturation at 98 °C for 30 s, denaturation at 94 °C for 5 s, annealing at 65 °C for 20 s, and extension at 68 °C for 20 min. Thirty cycles of amplification were followed by a final 10 min extension at 68 °C. The PCR product was recovered, purified with conventional methods, cloned with the pCR2.1-TOPO-Vector, and sequenced using a Big dye Terminator (ThermoFisher, Waltham, MA, USA).

### 4.6. Deletion Analysis of SLC22A5 Gene Promoter 5′ Terminus

The diagram of the deletion analysis of the SLC22A5 gene promoter 5′-terminus is shown in Figure 1. In accordance with the location of MEF2 binding sites, the design of the different lengths of the promoter was based on the template of the sequenced SLC22A5 gene 3256/+244 fragment.

### 4.7. Construction and Verification of Luciferase Reporter Gene Expression Plasmid of SLC22A5 Gene Promoter Fragments and MEF2-Binding Site Fragment

The different lengths of the SLC22A5 gene promoter were created by using restriction enzymes. Acc65I and Eco 32I, Eco32I and XbaI, Eco32I and XmajI, and Eco32 and Cfr10I double digestions were carried out on the SLC22A5 gene promoter-pCR2.1-TOPO construct. The fragments were separated and purified using preparative gel (1%) electrophoresis. The pGL3-basic was digested by Acc65I and SmaI, NheI, and SmaI and SmaI. The vectors and the target fragments were linked using a T4 ligase. Thereafter, competent DH5α cells were transformed, and positive clones were selected and verified through restriction endonucleases analysis and DNA sequencing. The successfully constructed plasmids were named pGL3-OCTN2 −3256/+244 (3500 bp), pGL3-OCTN2 −1962/+244 (2206 bp), pGL3-OCTN2 −592/+244 (836 bp), and pGL3-OCTN2 −54/+244 (298 bp).

The nucleotide sequences of the MEF2-binding site and mutated MEF2 binding site were created by Microsynth (MEF2 5′-TCACTAAGCCACTACTATTTTAAAAAGGAA-3′, mut MEF2 5′-TCACTAAGCCACTACTAGTTTAAAAAGGAA-3′). The pGL3-basic vector was double digested by using restriction enzymes Acc65I and XhoI. C2C12 cells were transfected with the construct of a pGL3-basic vector and Mef2-binding site inserts using LipofectaminTM 2000 according to the manufacturer’s instructions.

### 4.8. Transient Transfection of Reporter Gene Plasmids into Eukaryotic Cells

Murine skeletal muscle cells (C2C12 myoblasts) were grown on 12-well plastic dishes containing Dulbecco’s modified Eagle’s medium supplemented (DMEM) with 10% fetal bovine serum and 1% antibiotics (penicillin and streptomycin). Cells were grown to 80% confluence and the viability was assessed with a trypan blue exclusion before they were used in the experiments.

The reporter gene plasmid (experimental recombinant plasmid and empty vector control plasmid pGL3-enhancer) and equimolar internal control plasmid pRL-TK were co-transfected into C2C12 cells using LipofectaminTM 2000, according to the manufacturer’s instructions. Three wells were transfected for each plasmid. Transfection procedures were carried out following the operating instructions. The dosage of DNA was 1.6 μg/well. Cells were collected after 24 h of incubation.

For the treatment with the interferon-gamma and p-38-MAPK inhibitor (SB203580), the C2C12 blasts were transfected with the experimental recombinant plasmids and maintained in DMEM + 10% fetal bovine serum for 12 h, washed with phosphate-buffered saline, and incubated in a differentiation medium. Treatment with SB203580 (10 μM) or interferon-gamma (10 ng/mL) was performed for 12 h. The final concentration of DMSO was adjusted for each group (with or without reagents).

### 4.9. Dual-Luciferase Activity Assay

A dual-luciferase reporter Gene Assay Kit was used to assess the expression level of the reporter gene. The Relative Light Unit (RLU) was calculated as the ratio of the activity of the firefly luciferase to that of the Renilla luciferase. The dual-luciferase activity assay was conducted in accordance with the reporter gene analysis system operation manual. The efficiency of cells in each well was corrected with the activity of the internal control plasmid pGL3-RL.

### 4.10. Electrophoretic Mobility Shift Assay (EMSA) and Gel-Shift Competition Assay

Whole-cell lysates and nuclear extracts were prepared as described previously [40]. The lysates were centrifuged at 22,000× *g* for 5 min. Protein concentrations were determined with the Bio-Rad protein assay.

The antibody supershift assay is used to identify proteins present in a DNA-protein complex. For EMSA, nuclear (2 µL) extracts were incubated for 20 to 30 min at 20 °C in a mixture containing 20 mM HEPES (pH 7.9), 4% Ficoll, 1 mM MgCl2, 40 mM KCl, 0.1 mM EGTA, 0.5 mM DTT, and 160 µg of poly(dI-dC)-poly(dI-dC) per ml (Sigma) with 1 ng of 32P-labeled oligonucleotides. Samples were separated on a 4.5% nondenaturing polyacrylamide gel at 400 V for 2 h 40 min at 4 °C. Gels were then dried and exposed to BioMax MR (Kodak) films for 6 h to 1 day. The following oligonucleotides corresponding to MEF2 response element sequences were used: MEF2, 5′-GATTCACTAAGCCACTACTATTTTAAAAAGGAA-3′, and mut MEF2 5′-GATTCACTAAGCCACTACTAGTTTAAAAAGGAA-3′. For supershift experiments, 1 µL of 1:10-diluted antisera was added to the samples, as indicated.

The gel-shift competition assay is a technique used to assess the sequence specificity of a DNA–protein interaction, especially since most protein extracts contain both specific and nonspecific DNA-binding proteins. When one is assessing a specific DNA–protein interaction, an unlabeled competitor DNA probe with the same sequence as the protein binding site is used to sequester/compete for the protein away from the labeled probe. As a control for binding specificity, a nonspecific competitor probe must be used to show that the unlabeled nonspecific probe cannot sequester/compete for the protein away from the labeled probe. We used the unlabeled MEF2-probe as the competitor and the unlabeled mut MEF2-probe as a nonspecific competitor.

### 4.11. Immunoprecipitation and Immunoblotting

To measure the effect of IFNγ-stimulation on MEF2 and pGC1-α expression in C2C12 blasts, immunoprecipitation and immunoblotting were performed as described previously [41]. The intensity of each band was measured by densitometry analysis using NIH Image software (NIH, Bethesda, MD, USA).

### 4.12. Carnitine Transport into C2C12 Cells

Transport of ^3^H-carnitine into C2C12 cells overexpressing PGC-1α was performed in the presence of a buffer containing Na^+^ as described previously [42].

### 4.13. Quantitative Real-Time PCR

Total RNA was obtained from the gastrocnemius (40 mg), kidney (40 mg), and heart (40 mg) or from C2C12 cells overexpressing PGC-1α or green fluorescence protein (used as a control) using RNeasy mini kits (QIAGEN, Hilden, Germany) according to the manufacturer’s instructions. C2C12 cells overexpressing PGC-1α had been prepared by adenoviral transduction, as described previously [41]. Using reverse transcription, 0.5 µg of total RNA was converted into cDNA with an Omniscript RT kit (QIAGEN, Hilden, Germany). For the multiplication of target genes, cDNA was mixed with target gene forward and reverse primers (final concentration, 0.3 µM), and a fluorescent dye SYBR Green (Roche Diagnostics, Mannheim, Germany) was used for the measurement of duplex DNA formation during the PCR cycle in real-time. The real-time PCR cycling was performed with the ViiA™ 7 Real-Time PCR System (Applied Biosystems, Waltham, MA, USA) in triplicate. The sequence of the primer sets used were for the Slc22a5 forward primer 5′-TCC GAA CAC GGA ATA TCA GG-3′; the Slc22a5 reverse primer 5′-AGC CCA CTG ATA TGG TCA GC-3′; and for the Ppargc1a forward primer 5′-AAT GCA GCG GTC TTA GCA CT-3′ and Ppargc1a reverse primer 5′-ACG TCT TTG TGG CTT TTG CT-3′. Quantification of the gene expression was performed as described previously [43] using the 18s gene as the internal control using the forward primer 5′-AGT CCC TGC CCT TTG TAC ACA-3′; and 18s reverse primer 5′-CGA TCC GAG GGC CTC ACT A-3′. The quantification of Slc22a5 and Ppargc1a mRNA by real-time PCR in C2C12 cells was performed as described previously [44] with the same primers as described above.

### 4.14. Western Blotting

Around 50 mg of mouse quadriceps muscle were powdered with a Microdismembrator for 1 min at 3000 rpm (Sartorius Stedim Biotech, Aubagne, France) at a temperature of around −196 °C and later lysed on ice in a PhosphoSafe™ Extraction Reagent (Merck KGaA, Germany) at a 4 °C temperature. The homogenate was then centrifuged at 16,000× *g* at 4 °C for 10 min, and the supernatant was collected for further analysis. The protein content in the muscle homogenate supernatant was determined with a Pierce BCA protein assay kit (ThermoFisher Scientific, Waltham, MA, USA). For protein separation by size, 50 μg of protein was loaded onto the NuPAGE 4–12% Bis-Tris gel (ThermoFisher Scientific, Waltham, MA, USA). The electrophoresis was run at 140 V until full resolution, after which the gel was electroblotted to a nitrocellulose membrane (Bio-Rad, Hercules, CA, USA). Blotted proteins were probed against target proteins by using primary antibodies against β-actin (sc-47778, Santa Cruz Biotechnology, 1:1000), OCTN2 (ab180757, Abcam, 1:1000), and PGC-1α (sc-518025, Santa Cruz Biotechnology, 1:1000). After primary antibody overnight incubation, membranes were probed with secondary HRP=conjugated antibodies against rabbit (sc-2004, Santa Cruz Biotechnology, 1:2000) or mouse (sc-516102, Santa Cruz Biotechnology, 1:2000) antibodies for 1 h. Blotted protein bands were visualized with a chemiluminescent substrate (Clarity Western ECL substrate; Bio-Rad Laboratories, Hercules, CA, USA) in the Fusion Pulse TS device (Vilber Lourmat, Eberhardzell, Germany). Protein expression was quantified using the Evolution-Capt software version 17 (Vilber Lourmat, Eberhardzell, Germany). Western blots with C2C12 cells were performed as described previously [38] using the same antibodies as described above.

### 4.15. Animal Experiments

Two mouse models were used in this study: namely, mice with skeletal muscle PGC-1α overexpression (OE) and PGC-1α muscle knockout mice (KO), which were compared with wild-type mice (WT). Muscle-specific PGC-1α OE mice were generated using the DNA microinjection technique [27]. These mice express PGC-1α under the control of the muscle creatine kinase promoter, preferentially in type II fibers. The generation of muscle-specific PGC-1α knockout KO mice using the Cre/LoxP system by “floxing“ the Ppargc1a allele has been described previously [45]. Wild-type mice were from the same mouse line used for the generation of OE mice without PGC-1α overexpression. Mice were housed at 22°C ± 2 °C on a 12:12 h photoperiod and had free access to food and water. At the time point when the tissues were obtained, mice were 15–17 weeks old. The experiments performed agreed with the guidelines from Directive 2010/63/EU of the European Parliament on the protection of animals used for scientific purposes. Experiments had been reviewed and accepted by the Cantonal Veterinary Authority of Basel (License 2847).

### 4.16. Statistical Analysis

Each experiment was performed in triplicate, and the reported values represent the mean ± SD of three to five separate experiments. A student’s *t*-test was used for the comparison of two groups, and a one-way ANOVA was used for multi-group comparisons followed by the Holm–Sidak test to localize differences. *p* < 0.05 was considered statistically significant.

## Figures and Tables

**Figure 1 ijms-23-12304-f001:**
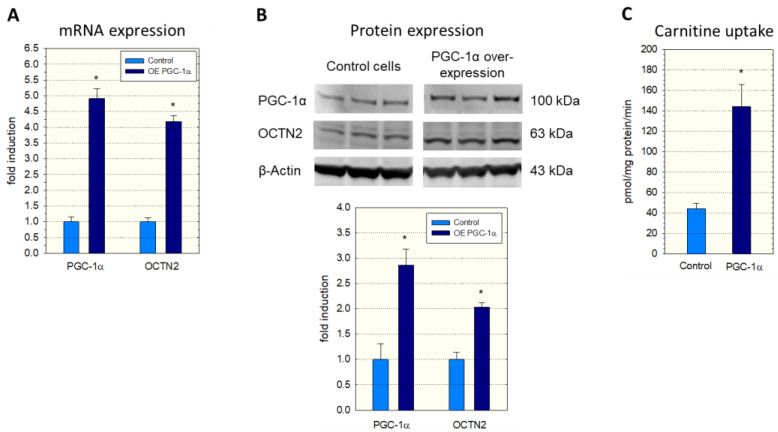
mRNA and protein expression of OCTN2 in and carnitine transport into C2C12 myoblasts with overexpression of PGC-1α. Overexpression of PGC-1α was achieved by adenoviral (**A**,**C**) or lentiviral transduction (**B**). (**A**) mRNA expression, (**B**) Western blot and quantification of the bands, and (**C**) Na^+^-dependent cellular uptake of ^3^H-carnitine. Control cells contained the respective empty vector. Data are presented as the mean ± SD of *n* = 3 (**A**,**B**) or 5 (**C**) independent observations. * *p* < 0.05 vs. control incubations.

**Figure 2 ijms-23-12304-f002:**
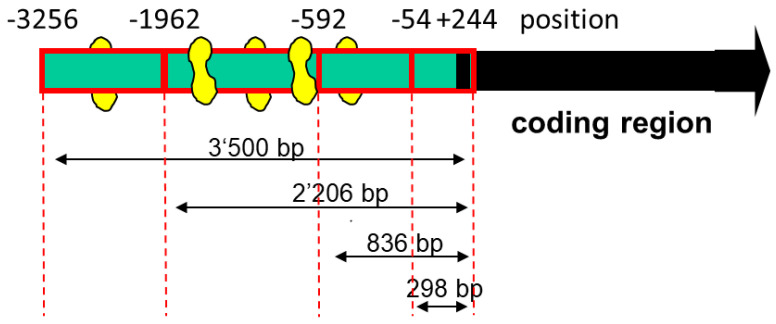
Diagram of the deletion analysis of the SLC22A5 gene promoter 5′ terminus. MEF2 binding sites are indicated in yellow. The design of the different lengths of the SLC22A5 gene promoter was based on the template of the sequenced −3256/+244 fragment. The −1624/−1595 fragment, which was used in the gel-shift competition and electrophoretic mobility shift assays, is not shown in the figure.

**Figure 3 ijms-23-12304-f003:**
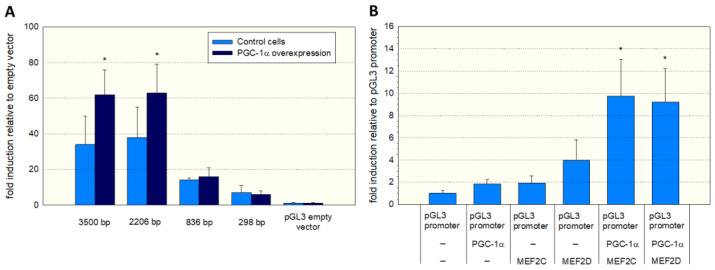
Functional analysis of the SLC22A5 gene promoter. (**A**) C2C12 myoblasts were transfected with different 5′-truncated fragments of the SLC22A5 gene 5′ upstream region and their functionality assessed using a dual luciferase assay. The −3256/+244 (3500 bp) and the −1962/+244 (2206 bp) promoter segments revealed the highest reporter gene activity. Co-transfection with PGC-1α increased the promoter activity of these segments. (**B**) C2C12 myoblasts containing the PGL3 promoter were transfected with MEF2C/D or PGC-1α and a combination of both. The highest promoter activity was achieved in C2C12 cells with combined transfection with PGC-1α and MEF2C or MEF2D. Data are presented as the mean ± SD of *n* = 5 independent observations. * *p* < 0.05 cells with PGC-1α overexpression vs. control cells without PGC-1α overexpression (**A**) and *p* < 0.05 vs. cells expressing the pGL3 promoter (**B**).

**Figure 4 ijms-23-12304-f004:**
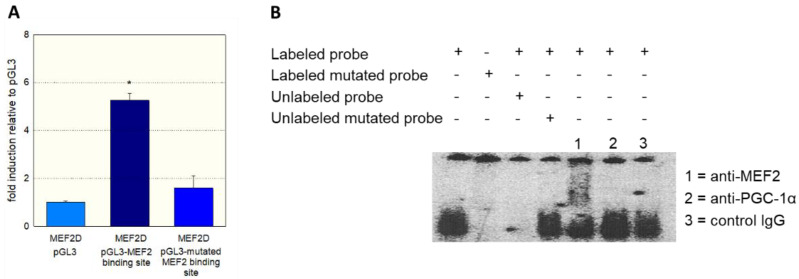
Interaction of MEF2 with the SLC22A5 gene promoter. (**A**) C2C12 myoblasts overexpressing MEF2D were transfected with the non-mutated −1624/−1595 fragment and with the −1624/−1595 fragment containing a T to G mutation in position −1607 of the SLC22A5 gene promoter. The mutation blunted the stimulation of the promoter activity by MEF2D. (**B**) Gel-shift competition and electrophoretic mobility shift (EMSA) assays using the unmutated and mutated labeled regulatory sequence −1624/−1595 proved the interaction of MEF2D with the SLC22A5 gene promoter. Data are presented as the mean ± SD of *n* = 3 independent observations. * *p* < 0.05 vs. pGL3.

**Figure 5 ijms-23-12304-f005:**
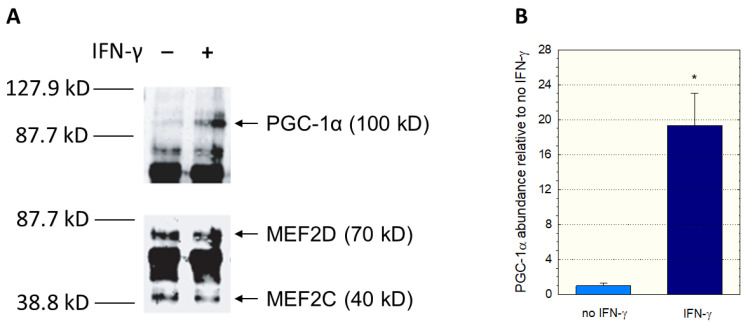
Interaction between PGC1α and MEF2. C2C12 myoblast lysates were incubated with anti-MEF2 antibodies, and the separated precipitates assayed using anti-MEF2 and anti-PGC-1α antibodies. The corresponding immunoblots are shown in (**A**) and the quantification of the PGC-1α band in (**B**). Treatment of C2C12 myoblasts with interferon-γ was performed for 12 h. Interferon-γ activates the p38 MAP kinase (p38 MAPK), which activates MEF2. Data are presented as the mean ± SD of *n* = 3 independent observations. * *p* < 0.05 vs. incubations without IFN-γ.

**Figure 6 ijms-23-12304-f006:**
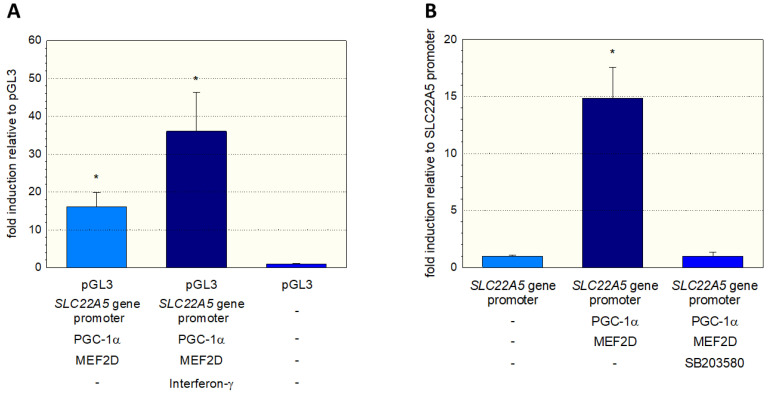
Interaction of the SLC22A5 gene promoter activity with p38 MAPK. C2C12 myoblasts were transfected with the SLC22A5 gene promoter, and the functionality assessed using a dual luciferase assay. (**A**) Interferon-γ, which activates p38 MAPK, increases the activity of the SLC22A5 gene promoter in C2C12 cells overexpressing PGC-1α and MEF2D. (**B**) The specific p38 MAPK inhibitor SB203580 blocks the stimulation of the SLC22A5 gene promoter activity. Data are presented as the mean ± SD of *n* = 5 independent observations. * *p* < 0.05 vs. pGL3 (empty vector).

**Figure 7 ijms-23-12304-f007:**
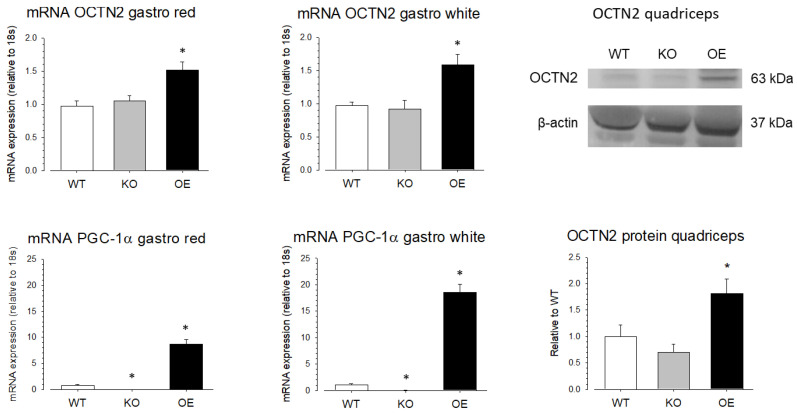
OCTN2 mRNA and protein expression in skeletal muscle of mice with muscular PGC-1α overexpression or knock-out. mRNA expression of OCTN2 and PGC-1α was determined by rt PCR in white and red gastrocnemius (gastro) of wild-type mice (WT), mice with muscular PGC-1α overexpression (OE), and mice with muscular PGC-1α knock-out (KO). Protein expression of OCTN2 was determined by Western blotting in the quadriceps of the same groups of mice. Data are presented as the mean ± SD of *n* = 4 independent observations. * *p* < 0.05 vs. WT mice.

## Data Availability

All original data are available on request.

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
