# Peer review of "PGC-1α and MEF2 Regulate the Transcription of the Carnitine Transporter OCTN2 Gene in C2C12 Cells and in Mouse Skeletal Muscle"

_ijms, 2022, doi:10.3390/ijms232012304_

Round 1

Reviewer 1 Report

The manuscript entitled “PGC-1α and MEF2 regulate the transcription of the organic cation transporter OCTN2 gene in C2C12 cells and in mouse skeletal muscle” describes the transcription factor MEF2 as a dimer partner of PCG-1alpha in the transcription regulation of OCTN2 in the skeletal muscle. Overall, the experiments seem to be properly designed and performed and the results support the initial working hypothesis. My specific comments/suggestions follow:

·         In figure 1, the authors show that the overexpression of PGC-1alpha indeed increase the expression of OCTN2. The increase in the mRNA level seems more evident than that at the protein level. A functional assay (i.e. Transport assay with [3H]carnitine in the presence or absence of Na+) should be added to better understand whether this induction has functional ramifications.

·         INFγ seem to promote PGC1alpha-MEF2 dimerization. What is the impact of INFγ treatment to OCTN2 expression and function in skeletal muscle cells? WB and transport experiments should be added to the gene reporter assay. This is important because INFγ has been shown to induce OCTN2 expression in other cell types (i.e. HEK293 and in Caco2BBE) but not in skeletal muscle cells. The following papers should be cited: PMID: 20722056, PMID: 29491466.

·         A minor point is the definition of OCTN2. The authors classify as a cation transporter. However, a carnitine/organic cation transporter is a more precise definition as carnitine exists as a zwitterion.

·         Another minor but important point is the use of the work promotor. In my mind, the correct word to define the region of DNA upstream of a gene where the transcription of that gene is initiated and/or regulated is promoter.

·         In the legend of figure 3 I found a typo. Precisely, “(3500 pb)”. I believe it is 3500 bp. Please check whether the typo has been repeated throughout the manuscript.

Author Response

  • In figure 1, the authors show that the overexpression of PGC-1alpha indeed increase the expression of OCTN2. The increase in the mRNA level seems more evident than that at the protein level. A functional assay (i.e. Transport assay with [3H]carnitine in the presence or absence of Na+) should be added to better understand whether this induction has functional ramifications.

Answer: we have determined the uptake of 3H-L-carnitine in the presence of a buffer containing Na+ as one of the first experiments after having observed that PGC-1α overexpression stimulates the expression of OCTN2 in C2C12 cells. We obtained the following results:

Control C2C12 cells (empty vector): 44.2±5.3 pmol/mg protein/min (average ± SD, n = 5 determinations)

C2C12 cells overexpressing PGC-1α: 144±22 pmol/mg protein/min (average ± SD, n = 5 determinations)

The difference corresponds to the increase of the Na+-dependent transport into C2C12 cells due to PGC-1α overexpression. The data are consistent with the mRNA and protein expression data shown in Fig. 1. We have included the data into the result section on page 2 and 3 and describe the analysis in the Method section on page 10.

  • INFγ seem to promote PGC1alpha-MEF2 dimerization. What is the impact of INFγ treatment to OCTN2 expression and function in skeletal muscle cells? WB and transport experiments should be added to the gene reporter assay. This is important because INFγ has been shown to induce OCTN2 expression in other cell types (i.e. HEK293 and in Caco2BBE) but not in skeletal muscle cells. The following papers should be cited: PMID: 20722056, PMID: 29491466.

Answer: Thank you for this comment and for suggesting the citation of additional references regarding the effect of INFγ on the expression of OCTN2. In our experiments, we used INFγ as a stimulator of the p38 mitogen-activated protein kinase (p38 MAPK), which has been shown to phosphorylate and thereby activate MEF2 (Han J et al. Nature 1997;386:296-299). The immunoprecipitation experiment in Fig. 5 is consistent with activation of MEF2 by INFγ and suggests that the activated MEF2 recruits more PGC-1α. The role of p38 MAPK in the activation of the OCTN2 promotor (and indirectly of MEF2) is shown in Fig. 6B. INFγ/MEF2/PGC-1α activate the OCTN2 promotor only efficiently when the p38 MAPK signaling pathway is functioning.

While our experiments suggest that INFγ is functioning via the stimulation of the p38 MAPK pathway in C2C12 cells, we cannot exclude the possibility that other mechanisms are involved in the stimulation of OCTN2 expression by INFγ. The studies in HEK293 and Caco2BBE cells show that INFγ can increase OCTN2 expression and function also in other cells than C2C12 cells, but the mechanisms need to be elucidated.

We agree that functional and protein expression studies would have been more convincing to show the effect of INFγ on OCTN2 expression than the provided immunoprecipitation and reporter gene experiments. However, the short period of time given to answer the questions of the Reviewers did not allow us to perform new experiments involving cell culture. We regarded the experiments involving INFγ as a possibility to show the involvement of the p38 MAPK/MEF2 pathway in increasing OCTN2 expression (together with the inhibition study using SB203580). This is the reason why we did not assess OCTN2 expression and/or function in C2C12 cells treated with INFγ.

We have included the papers suggested by the Reviewer and have modified the discussion regarding the possible role of INFγ, p38 MAPK and MEF2 in the regulation of the expression of OCTN2 in C2C12 cells on page 7.

  • A minor point is the definition of OCTN2. The authors classify as a cation transporter. However, a carnitine/organic cation transporter is a more precise definition as carnitine exists as a zwitterion.

Answer: we agree with the Reviewer and have replaced cation transporter by carnitine transporter.

  • Another minor but important point is the use of the work promotor. In my mind, the correct word to define the region of DNA upstream of a gene where the transcription of that gene is initiated and/or regulated is promoter.

Answer: thank you for correcting us, you are of course right. We have replaced promotor by promoter in the entire manuscript including the figures.

  • In the legend of figure 3 I found a typo. Precisely, “(3500 pb)”. I believe it is 3500 bp. Please check whether the typo has been repeated throughout the manuscript.

Answer: Thank you, we have corrected the mistake and checked the manuscript.

Reviewer 2 Report

The findings provided by Katerina Novakova et al. to demonstrate a novel mechanism by which MEF2 and PGC-1α regulate the transcription of the OCTN2 gene in skeletal muscle are novel. OCTN2 being targeted by multiple transcriptional factors supports the essential role of OCTN2 in muscle metabolism. Minor changes can improve the manuscript further.

1)      Figure 1A shows the usage of plasmid with GFP tag for control conditions. Based on the molecular weight of PGC1α in Fig. 1B and the plasmid information provided in the methods section, the PGC-1α construct does not contain any GFP tag. Comparing GFP and non-GFP conditions is not ideal, as GFP by itself can influence several factors including cytotoxicity and immunogenicity as reported earlier in several publications.

2)      The whole work tried finding an alternative transcriptional signaling pathway involving PGC-1α that regulates Octn2 expression as PPAR-α knock-out mice did not show a reduced Octn2 expression in skeletal muscle, as explained in the first two lanes of the discussion. After all, when PGC-1α knock-out mice didn’t show any change in expression of Octn2, authors re-proposed the involvement of alternative factors, including PPAR-α, as discussed in lanes 242 and 243.  Authors should reconsider the explanation for this part of the discussion.

3)      The time of Interferon-gamma treatment have discrepancies. Lane 359 in methods section mentioned it as 48 hours. Lane 169 in results section it is mentioned as 12 hours.

4)      Details on the sequence of SLC22A5 promoter used in figure 6 are lacking. Figure 6B shows that PGC-1α overexpression induced the SLC22A5 promoter activity by 15 times upon comparing with the condition without PGC-1α OE or with an empty vector. On the other hand, data provided in Figures 3A and 3B shows that the effect of PGC-1α overexpression on the promoter activity of different deletion SLC22A5 constructs, including the complete promoter construct (3500 bp), did not exceed by two times. The differences in findings should be clarified with information on the promoter used in figure 6.

5)      Fig 4A should be labelled with MEF2D overexpression.

6)      Western blots performed using Co-IP extracts showed a non-specific band both in PGC-1α and MEF2 blots. While demonstrating the interaction between PGC-1α and MEF2 in Co-IP experiments, the usage of appropriate control conditions should be considered. For example, control conditions of incubation of cell extract with control IgG antibodies followed by blotting with PGC-1α and MEF2 antibodies.

7)      MEF2 antibody detected two bands in IP extracts. Based on molecular weight, one can assume the 70 kDa band is the MEF2D isoform, and the 40 kDa band is the MEF2C isoform. If true, one should consider replacing molecular weights with isoform names consistent with earlier figures. Catalog information of MEF2 antibody from Santa Cruz that detects both isoforms should be provided in the methods section.

Author Response

1) Figure 1A shows the usage of plasmid with GFP tag for control conditions. Based on the molecular weight of PGC1α in Fig. 1B and the plasmid information provided in the methods section, the PGC-1α construct does not contain any GFP tag. Comparing GFP and non-GFP conditions is not ideal, as GFP by itself can influence several factors including cytotoxicity and immunogenicity as reported earlier in several publications.

Answer: The reviewer is correct, the control cells shown in Fig. 1A were transfected with a plasmid containing gfp. Since we also have data with the empty vector, we replaced the current control cells with those transfected with the empty vector. The interpretation of the data does not change. The carnitine uptake experiments (point 1 of Reviewer 1) have been conducted with control cells containing the empty vector.

2) The whole work tried finding an alternative transcriptional signaling pathway involving PGC-1α that regulates Octn2 expression as PPAR-α knock-out mice did not show a reduced Octn2 expression in skeletal muscle, as explained in the first two lanes of the discussion. After all, when PGC-1α knock-out mice didn’t show any change in expression of Octn2, authors re-proposed the involvement of alternative factors, including PPAR-α, as discussed in lanes 242 and 243.  Authors should reconsider the explanation for this part of the discussion.

Answer: as shown in Fig. 7, PGC-1α knock-out was associated with an only numerical decrease in the mRNA and protein expression of OCTN2 in mouse skeletal muscle. In the paper of van Vlies et al. (BBA 2007;1767:1134-1142), muscle OCTN2 mRNA expression was not decreased in PPAR-α knock-out mice. It, therefore, appears that OCTN2 mRNA expression can be maintained in the absence of PPAR-α (possibly by PGC-1α/MEF2) and in the absence of PGC-1α (possibly by MEF2 and PPAR-α). We have corrected the statement in the discussion accordingly (page 7).

3) The time of Interferon-gamma treatment have discrepancies. Lane 359 in methods section mentioned it as 48 hours. Lane 169 in results section it is mentioned as 12 hours.

Answer: Treatment with interferon gamma or SB203580 was for 12 h. We have corrected that on page 10.

4) Details on the sequence of SLC22A5 promoter used in figure 6 are lacking. Figure 6B shows that PGC-1α overexpression induced the SLC22A5 promoter activity by 15 times upon comparing with the condition without PGC-1α OE or with an empty vector. On the other hand, data provided in Figures 3A and 3B shows that the effect of PGC-1α overexpression on the promoter activity of different deletion SLC22A5 constructs, including the complete promoter construct (3500 bp), did not exceed by two times. The differences in findings should be clarified with information on the promoter used in figure 6.

Answer: We agree with the Reviewer. In the experiments shown in Figure 6, the entire SLC22A5 promotor was used as shown in Fig. 2. Figure 3B and 6B are therefore comparable. However, in the experiments shown in Fig. 6, C2C12 cells not only overexpressed PGC-1α, but also MEF2D. We have changed that in the figure labels and in the description of the figure. We apologize for this mistake in the original version of the manuscript. The stimulation of the promotor activity can now be explained by the presence of MEF2 and INF-γ. The results shown in Fig. 6B are also consistent with those in Fig. 6A.

5) Fig 4A should be labelled with MEF2D overexpression.

Answer: This is correct. We have stated MEF2D overexpression in the legend to the figure and have included it also in the text of the Result section on page 4. We also included it in the figure label, as requested.

6) Western blots performed using Co-IP extracts showed a non-specific band both in PGC-1α and MEF2 blots. While demonstrating the interaction between PGC-1α and MEF2 in Co-IP experiments, the usage of appropriate control conditions should be considered. For example, control conditions of incubation of cell extract with control IgG antibodies followed by blotting with PGC-1α and MEF2 antibodies.

Answer: We agree with the Reviewer that the use of a control IgG antibody would have been important to show the origin of non-specific bands. Unfortunately, we did not conduct such a control experiment. From other studies using the same cell system, we know that the large band at approximately 50 kD corresponds to the heavy chain of the IgG. Unfortunately, the time given for writing the rebuttal was too short to perform the requested experiments.

7) MEF2 antibody detected two bands in IP extracts. Based on molecular weight, one can assume the 70 kDa band is the MEF2D isoform, and the 40 kDa band is the MEF2C isoform. If true, one should consider replacing molecular weights with isoform names consistent with earlier figures. Catalog information of MEF2 antibody from Santa Cruz that detects both isoforms should be provided in the methods section.

Answer: Thank you for this remark. This is of course correct. We have labelled Fig. 5A accordingly.

Round 2

Reviewer 1 Report

Thanks to the authors to reply to all my comments. I do not understand why the new transport data are not supported by a figure. It can be easily added to figure 1.

Author Response

Thank you for your second review. According to your request, we have prepared a new figure 1 and have included the carnitine transport data. The reason, why we did not include the data into Fig. 1 in our first rebuttal was the fact that we measured only the sodium-dependent transport and not the sodium-independent transport. Nevertheless, the difference between PGC-1alpha overexpression and control (empty vector) corresponds to the sodium-dependent increase in in carnitine transport due to PGC-1alpha overexpression.